# Thermal Recycling of Glass Fibre Composites: A Circular Economy Approach

Maria Iglésias [1], Paulo Santos [2] , Tânia M. Lima [2,*] and Laura Leite [1]

1 Department of Electromechanical Engineering, University of Beira Interior, 6201-001 Covilhã, Portugal
2 C-MAST—Centre for Mechanical and Aerospace Science and Technologies, University of Beira Interior, 6201-001 Covilhã, Portugal
* Correspondence: tmlima@ubi.pt

**Abstract:** Composite materials are used in a wide range of applications, but due to their inherent nature of heterogeneity, particularly for thermoset-based polymer composites, their recycling is a problem, and their life cycle management remains one, too. This study applies a circular economy approach to the problem of excess waste of glass fibre (GF) composites and seeks a solution by testing a methodology for thermal recycling of GF composites by combining different times and temperatures. Through the by-hand lay-up process, diverse laminates were manufactured with recycled GF, and the mechanical results were compared with those of the control laminate; in this way, we sought to reinsert recycled fibres into a new life cycle, closing the loop of the material. The static properties, tensile tests, and three-point bending (3PB) tests were studied as well as the viscoelastic behaviour of the recycled fibres and respective laminates. For woven fibres, we highlight the recycling process at 600 °C for 15 min, which revealed a loss of only 15.3% of the tensile strength. The laminates with fibres recycled at 400 °C for 180 min presented a reduction of 52.14% and 33.98% for tensile and flexural strength, respectively, representing the best solution. For all laminates, the bending stress, stiffness, and strain are sensitive to the strain rate, and the tendency observed for these properties can be supported by linear models. Subsequently, for the best results, the stress–relaxation and creep behaviour were analysed, and it is possible to conclude that temperature and time of fibre recycling influence the viscoelastic response of laminates.

**Keywords:** circular economy; glass fibre composites; thermal recycling; mechanical characterisation; viscoelastic behaviour





## 1. Introduction

Increasingly, the environment and the impact of human life on Earth are topics discussed in the political sphere. This is due to the current climate crisis resulting from excessive human-caused pollution and mismanagement of natural resources. This climate crisis has increased society's concern and its perception of the climate emergency in aspects such as the long-term effects of current actions, the consumption of fossil fuels, and the energy crisis that can result from this dependence but also from the growing amount of pollution resulting from human activity, among others [1,2].

This has led to changes in the perception and habits of communities, which consequently have led to the development of strategies that incorporate the concepts of sustainable development, which seek to achieve a balance between needs and the surrounding environment These greener strategies may include the adoption of alternatives to fossil fuels or the creation of more opportunities for using secondary materials, such as recycled composites, and as part of their adoption, efforts have been made to define indicators and frameworks that are able to quantify the circularity associated with processes and products, with the European Commission defining some within the scope of the Action Plan for the Circular Economy [3].

As a result of this interest on the part of the European Commission, a set of indicators that monitor the involvement of countries in these initiatives was published, considering different themes and categories [4]. One of the most outstanding indicators is the rate of materials that are the result of a circular activity and that result from recycling processes, included in the category of secondary raw materials. This reflects the global use of recycled material that is used again, replacing virgin raw material [4]. Although this value has increased significantly in recent years, in 2020, at the European level, the rate for secondary materials was 12.8%, while in Portugal, it was only 2.2% [4]. This low value reflects the barriers that these materials have to overcome. These are mainly due to the lack of confidence that exists in this type of materials, as there is no legislation that regulates quality standards and the difference between transport and the cost associated with it [5].

Thus, as a means of resolving the low rate of circularity of materials both at the national and European levels, the European Commission has defined goals, among which is the importance of developing and standardizing procedures that translate into the design of more ecological products, considering their quality, durability, and impact throughout their useful life [6]. This concept of useful life relies on the concept of the life cycle, meaning a set of consecutive and interconnected stages of a product from the generation or acquisition of the raw material that constitutes it to its final disposal [7]. However, nowadays, the end of the life cycle is not always considered when designing products, which translates into a lack of attention regarding the different possibilities besides landfilling. That could result in a gain of value when considering a circular approach, such as the application of recycled composites as secondary materials, and a decrease in their environmental impact, as landfilling represents the worst option when compared to recycling [8,9].

However, recycling remains a challenge not only because of the properties of the materials but also because of the low cost of their manufacturing. To make the recycling process profitable and accessible, there must be strict control in this process, making it economically profitable [10].

Recycling comprises several processes, one of which is the extraction of glass fibre from the composite material. Likewise, it is possible to use the shredding of the composite to reuse the solid waste [11]. Today, mainly focusing on reinforcing fibres' design, manufacturing, and end-of-life management, three recycling methods are currently being developed: mechanical, chemical, and thermal [12].

Mechanical recycling starts with cutting, followed by the crushing of the materials, from which can be obtained filling particles or fibre particles. On the other hand, thermal recycling can be through a combustion process as well as fluidised bed combustion, pyrolysis, and microwave-assisted pyrolysis. Simply put, thermal recycling involves decomposing the different materials through external heating. Finally, chemical recycling consists of the dissolution of a solid compound using different solvents. There are two types of chemical recycling: solvolysis that uses organic solvents and hydrolysis that uses water as a solvent [13,14].

These three groups of methodologies of recycling can be considered when choosing the most appropriate for recycling glass fibre composites. Considering their good properties for certain applications in areas such as transport, construction, and renewable energies infrastructures, these materials have been growing exponentially. For example, the glass fibre composites' ability to mould itself to complex shapes, ability to be produced from a very small amount of raw material, easy acquisition, and being an extremely inexpensive and lightweight material have made them one of the most versatile materials when considering the composite world. Glass fibres exhibit excellent and useful bulk properties such as hardness, transparency, resistance to chemical attack, stability, and inertness in addition to desirable fibre properties such as strength, flexibility, and stiffness [15].

In recent years, several studies have been carried out on the possibilities of reuse of materials, from their burning to produce energy to the different types of recycling that could be applied to materials as well as the receptivity for the use of these new materials that can lose up to 60% of their initial characteristics [16,17]. Since glass fibre represents

a major source of silica, it is recurrently incorporated into concrete mixtures after undergoing mechanical recycling [18]. However, although it represents a good solution also for the reduction of the ecological footprint associated with the cement and concrete industry, this approach to the reuse of the material does not take advantage of its excellent mechanical characteristics and only generates products with a low added value. Therefore, several authors have been developing studies on the possibility of using fibres that result from thermal or chemical recycling although many of the processes that translate into a better use of fibres present great a great challenge in their expansion to the industrial scale [16].

Krauklis et al. [8] sought to identify different solutions to the problems of the end-of-life cycle of wind turbines, focusing on the reuse of sectioned blades and their recycling. They highlighted some of the recycled fibres, namely with regard to the loss of mechanical characteristics greater than 50% compared to virgin fibre. Dorigato [16] compared the various types of recycling methodologies, focusing his assessment on mechanical recycling and the combined method of thermochemical recycling and trying to explain how they are processed and what their benefits are as well as discussing how different parts of the wind turbine are processed at the end of the cycle of life. Rahimizadeh et al. [19] compared the tensile properties of fibres recovered through mechanical and thermal processes. Although the fibre tensile test presented better results when considering the mechanically recycled fibres, considering the resulting stiffness of the fibres, the ones recycled thermally represented better results.

A well-known problem with thermal recycling, taking place usually at between 450 to 600 °C [20–24], of waste glass fibre composites is the weakening of the fibre reinforcement. Large reductions to fracture stress limit the use of recycled glass fibres in new structural composite materials. Heat treatment of glass fibres reveals a rapid reduction in fracture stress with increasing temperatures above 250 °C, and the fibre strength is reduced by 55 to 70% [20–23].

It is known that composite materials are highly sensitive to the strain rate, and in this study, considering only static properties reveals to be very a conservative method [25,26]. Furthermore, when they are used in long-term applications, as a consequence of the inherent viscoelasticity of the matrix phase, composites are prone to creep and stress relaxation even at room temperature [26–28].

Considering the problem associated with the prohibition or legislation for limitation of the quantity of materials that go to landfill when their useful life ends, our main goal is to study the applicability of the thermal recycling of glass fibre composites and the reintegration of the recycled fibres in new laminates [8,9].

Thus, with the goal of continuing the studies that have already been developed on the recycling of glass fibres, we sought to identify a methodology and new opportunities for the application of recycled fibres so that their life cycle has a more sustainable approach and is more geared towards a circular economy strategy, and herein, we try to present a solution that would both benefit the environment but also the companies that produce composite waste [9].

## 2. Materials and Methods

Nine equal laminates with dimensions $290 \times 165 \times 1.2$ mm were manufactured, each one with eight layers of type E glass-fibre-woven bidirectional fabric (taffeta with 195 g/m$^2$), and were combined with a Sicomin SR 8100 resin and a SD 8824 hardener, both supplied by Sicomin, in a weight ratio of resin to hardener 100:22. One by one, the glass fibre layers were stacked, taking care to align them and evenly distribute the resin by hand lay-up technique. Then, the assembly was then placed inside a vacuum bag, heat sealed, and raised to approximately $-0.9 \pm 0.1$ bar. This step was intended to extract the maximum amount of small air bubbles that, due to the nature of the manufacturing procedure applied to the glass fibre fabric mesh, will be retained between the various layers of fibre and homogenise the distribution of the resin. A load of 2.5 kN was applied for 24 h to maintain a constant fibre volume fraction and a uniform laminate thickness. During the first 4 h,

the bag remained attached to a vacuum pump to eliminate any air bubbles existing in the composite. Finally, the post cure was followed according to manufacturer datasheet in an oven at $40 \pm 2$ °C for 24 h. After the manufacturing process was completed, the thermal recycling stage of the laminates was carried out. The laminates described above were submitted to recycling, the temperature (°C) and time (min) of which are duly indicated in Table 1 and completed in a TermoLab oven.

**Table 1.** Time and temperature combinations of recycling.

| Temperature (°C) | Time of Recycling (min) | | | | | |
|---|---|---|---|---|---|---|
| 400 | 15 | 30 | 60 | 120 | 180 | 240 |
| 600 | 15 | 30 | 60 | | | |

Other combinations of temperature and time of recycling were tested although they did not present themselves as scientifically relevant to this study because of the lack of good properties and fragility of the fibre after being recycled. Therefore, they were not considered for this study.

### 2.1. Mechanical Characterisation of Threads Woven of Recycled Glass Fibre

Mechanical characterisation of the glass fibres obtained by this recycling method was carried out using two layers of recycled composite. The tensile tests were carried out on the glass fibre strands as well as on the glass fibre recycled at room temperature and according to the procedure described in ASTM D578 and ASTM D579, respectively.

The glass fibre threads were removed from the differently woven ones carefully so as not to damage them; however, due to the impossibility of extracting them all perfectly, it was not possible to characterise all and compare the results with the control glass fibre thread (as a result of the excessive presence of resin for the low recycling times or its degradation at high temperatures). To obtain repeatability, six tests for each condition were considered. Specimens of woven fibre with a length of $115 \pm 0.2$ mm and a width of $10 \pm 0.2$ mm were produced so that they all contain nine fibre strands in their width (Figure 1a).

In the case of thread traction tests, these were performed on a universal machine, Adamel Lhormargy, model DY 35, equipped with a 100 N load cell and monitored by the AutoTrac software. These tests were carried out at room temperature, with a distance between grips of 150 mm and a deformation speed of 150 mm/min. The tensile tests of the woven glass fibre were performed on a Shimadzu universal machine, model AGS-X, equipped with a 10 kN load cell and the Trapezium X software, version 1.4.0, to acquire the results, and for each condition, at least five samples were tested at room temperature.

To obtain a better visualisation and characterisation of the different samples of fibres, we used a scanning electron microscope (SEM) of the Hitachi brand, model S-3400N. Finally, a quantitative analysis of chemical elements present in the samples was carried out using an energy dispersive X-ray (EDX) spectrometry detector from Bruker, model 5010, with a resolution of 129 eV in K$\alpha$ of Mn. This equipment has Quantax ESPRIT data acquisition and analysis software.

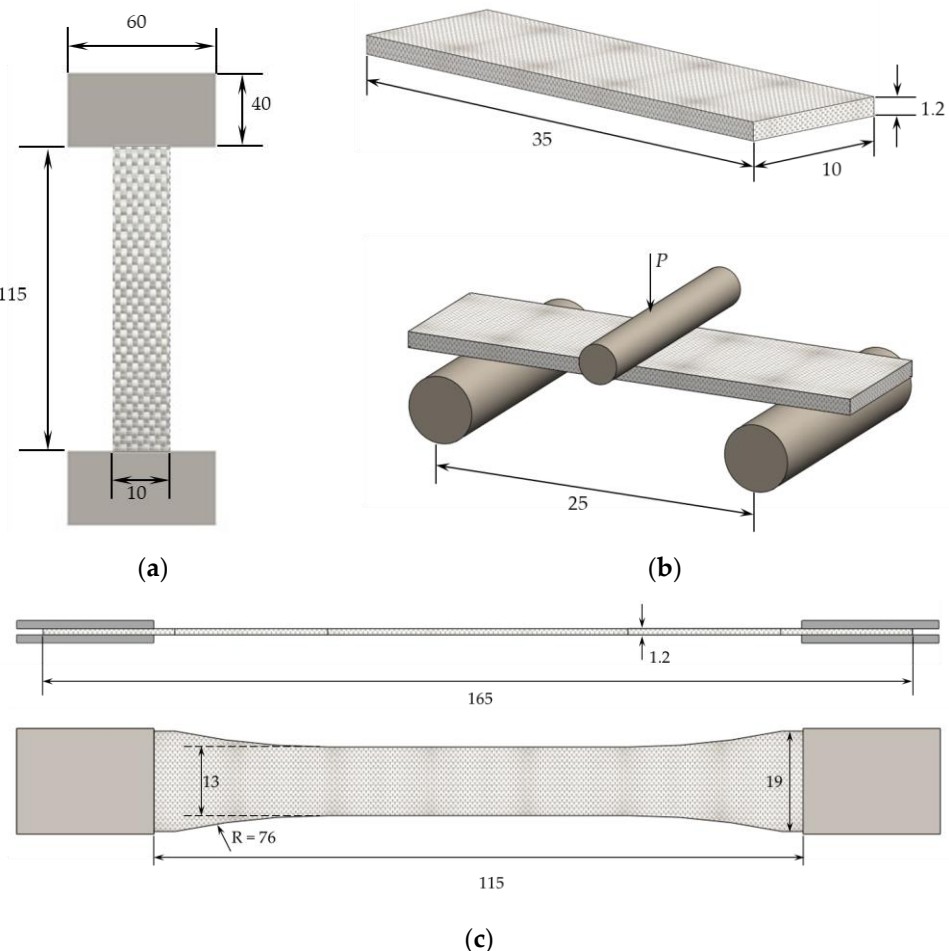

**Figure 1.** Geometry and dimensions of specimens: (**a**) glass fibres after thermal recycling; (**b**) 3PB tests and apparatus tests; (**c**) tensile tests. All dimensions in mm.

### 2.2. Static Characterisation of Laminates of Recycled Glass Fibre

After the recycling process, to evaluate the properties of the fibres as a reinforcement of composites, a glass fibre control laminate and nine new laminates with those same fibres were then produced, with six layers of recycled fibre fabric, following the entire procedure of manufacture described above and with the same dimensions: an average thickness of 1.2 mm and a fibre volume fraction of the glass between 41.9% to 46.1%.

Before manufacturing the laminates, it was necessary to clean the fibres, trying to eliminate small, loose residues of ash and debris of different sizes that were retained between the various layers and on the external surfaces of the laminates. Although it was not possible to carry out a cleaning in its entirety due to the nature of the various impurities embedded in the fibres and their fragility, a manual cleaning was carried out with a brush so as not to damage the fibre and not add costs to the process, followed by a visual inspection.

Subsequently, the laminates were cut into specimens. To guarantee the reliability of the results, at least three results were considered for tensile, and at least five results were considered for the flexural tests and viscoelastic behaviour.

In the case of tensile tests, the samples were cut in a Pronum Water Jet waterjet cutting machine to characterise the different laminates when subjected to a tensile stress. The dimensions associated with these specimens followed the indications of the ASTM International standard D638-14, according to Figure 1c, and were subjected to a tensile stress of 2 mm/min.

The tensile tests were performed on a Shimadzu universal machine, model AGS-X, equipped with a 100 kN load cell and aided by a strain gauge for precise characterisation of the different laminates. In these tests, the tension was calculated considering the expression described below.

$$\sigma = \frac{P}{A} \tag{1}$$

The value of the area is that of the cross-section of the specimen, given by $A = b \times h$. The rigidity of the specimens was calculated through linear regression of Hooke's law, always with a correlation factor greater than 95%.

$$\sigma = E \times \varepsilon \tag{2}$$

The specimens tested in the three-point bending (3PB) tests for static and viscoelastic characterisation were cut with the aid of a Struers Accutom 2 cutting machine, using a diamond disk and water-cooled to avoid the heating of the composite considering the advance speed. The dimensions associated with these specimens were defined based on the BS EN ISO 178:2003 standard, which defines a method to determine the flexural properties of a thermoset, with a span of 25 mm, as shown in Figure 1b.

The flexural strength was calculated as the rated stress in the mid-span section, obtained using the maximum load value and by applying the Equation (3):

$$\sigma = \frac{3\,P\,L}{2\,b\,h^2} \tag{3}$$

where $P$ is the load, $L$ is the distance between supports, $b$ is the width, and $h$ is the thickness of the specimen. The stiffness modulus was calculated using the linear elastic bending beam theory relationship:

$$E = \frac{\Delta P\,L^3}{48\,\Delta u\,I} \tag{4}$$

Here, $I$ is the moment of inertia of the cross-section, and $\Delta P$ and $\Delta u$ are, respectively, the load change and the mid-span bending displacement change for an interval in the linear region of the load versus displacement graph. The stiffness modulus was obtained by linear regression of load–displacement curves considering the interval in the linear segment with a correlation factor greater than 95%.

### 2.3. Viscoelastic Behaviour of Laminates of Recycled Glass Fibre

The bending strain was calculated by the following equation:

$$\varepsilon_f = \frac{6\,S\,h}{L^2} \tag{5}$$

Here, $S$ is the deflection, $L$ is the span length, and $h$ is the thickness of the specimen.

Regarding the strain rate, the tests were carried out over a range of different displacement rates, i.e., 200, 20, 2, 0.2 and 0.02 mm/min, and the bending properties (modulus, strength, and failure strain) were determined as a function of the strain rate, according to Equation (6):

$$\dot{\varepsilon} = \frac{d\varepsilon_f}{dt} = \frac{6\,V_T\,h}{L^2} \tag{6}$$

where $\varepsilon_f$ is the peripheral strain of the fibre, $t$ is the time, $V_T$ is the velocity of the third point, $L$ is the distance between supports, and $h$ is the thickness of the specimen. Strain rates ($\dot{\varepsilon}$) of $2.54 \times 10^{-1}$, $2.54 \times 10^{-2}$, $2.54 \times 10^{-3}$, $2.54 \times 10^{-4}$, and $2.54 \times 10^{-5}$ s$^{-1}$ for laminates were obtained according to Equation (6).

In the tests of stress relaxation and creep, in order to guarantee that these are carried out still in the elastic regime, the loads chosen for their performance correspond to 50% of the maximum stress verified in the 3PB tests at a deformation speed of 2 mm/min. Due to

the characteristics of the materials, it was decided to only carry out these tests on materials that, within their temperature, presented the best characteristics. Thus, only two different stress–relaxation and creep behaviours of recycled fibre laminates are presented and compared with the control laminate.

## 3. Results

### 3.1. Characterisation of Recycled Glass Fibre Strands and Woven Threads

After recycling, strands were removed from each of the woven threads, which were subsequently subjected to tensile tests in order to obtain their mechanical strength. For all conditions, the curves show a linear behaviour up to the maximum load followed by fibre collapse (Figure 2a). According to Rahimizadeh et al. [19], this is the typical behaviour observed for E-type glass fibres. Figure 2b summarises the results obtained in the strand tensile tests in terms of average values (marks), and the bands represent, respectively, the maximum and minimum values for each condition in terms of time and temperature and compared these values with the value of the control fibre. For both fibre recycling temperatures and looking at Figure 2b, it is observed that as the recycling time increases, the load value decreases.

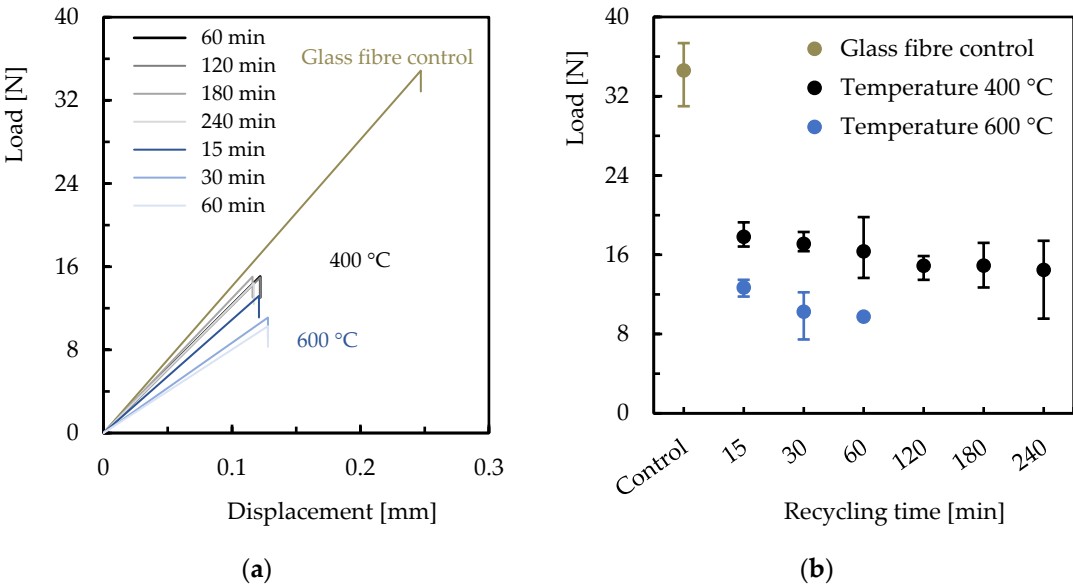

**Figure 2.** Mechanical characterisation of tensile strands subjected to temperatures of 400 °C and 600 °C: (**a**) representative load–displacement curves for glass fibre and recycled fibres; (**b**) results obtained from recycled glass fibre strand samples.

Regarding woven thread characterisation, Figure 3 shows the behaviour and load values obtained from samples recycled at temperatures of 400 °C and 600 °C and at different times compared to the control woven thread curve.

From Figure 3a, it is possible to observe, for control fibre fabric, a linear increase of the load strain with the displacement, followed by a non-linear behaviour in which the maximum load is reached. Regarding reclaimed fibre fabrics, they mostly present behaviour with a linear region up to the maximum load, followed by a significant drop after this value. Figure 3b summarises the results obtained in the woven tensile tests in terms of average values (marks) and the maximum and minimum values (bands) for each of the various conditions.

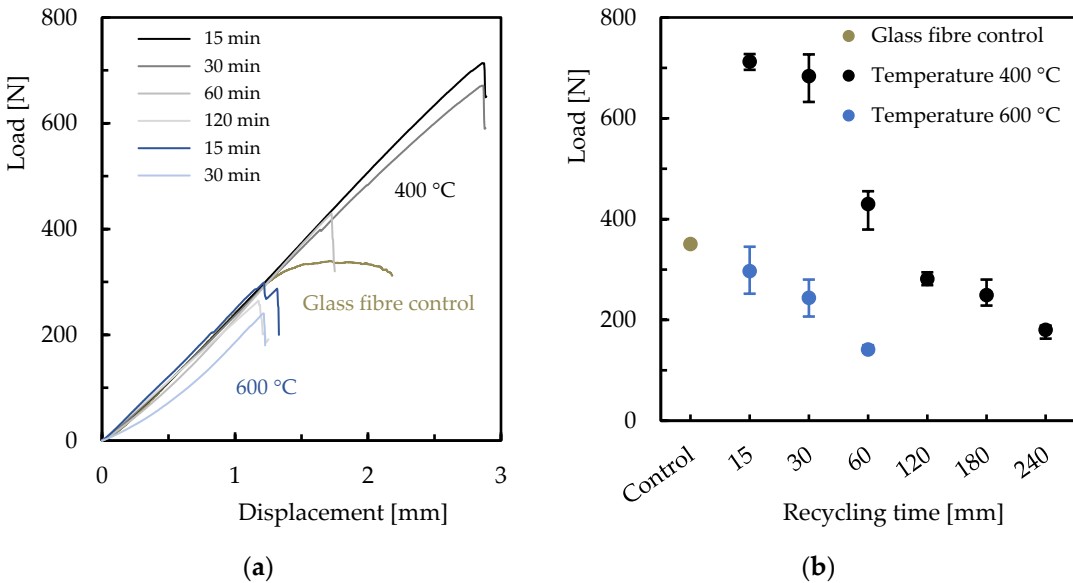

(**a**)  (**b**)

**Figure 3.** Mechanical characterisation of the tensile woven glass fibres subjected to temperatures of 400 °C and 600 °C: (**a**) representative load–displacement curves for woven glass fibres and recycled woven fibres; (**b**) results obtained from recycled woven glass fibre samples.

By observing Figure 3b, it can be concluded that recycling glass fibres at a lower temperature (400 °C) promotes better results compared to a higher temperature (600 °C). For example, for a time of 15 min, the value of the load at 400 °C is 58.3% higher in relation to the temperature of 600 °C. Regarding the control woven fibre, for the 400 °C temperature, the reference value is for the time of 60 min, while for the 600 °C temperature, it is 15 min.

### 3.2. Tensile Properties

For all laminates, tensile tests were carried out to evaluate the effect of temperature and time. The following Figure 4 shows representative samples for the two temperatures studied as well as some recycling times and compares them to the control sample. Regarding the data mode, it is very similar between all of them. It should be noted that laminates manufactured with fibres exposed for a longer time to a certain temperature show brittle damage due to the degradation of the fibres and poor fibre/epoxy resin adhesion.

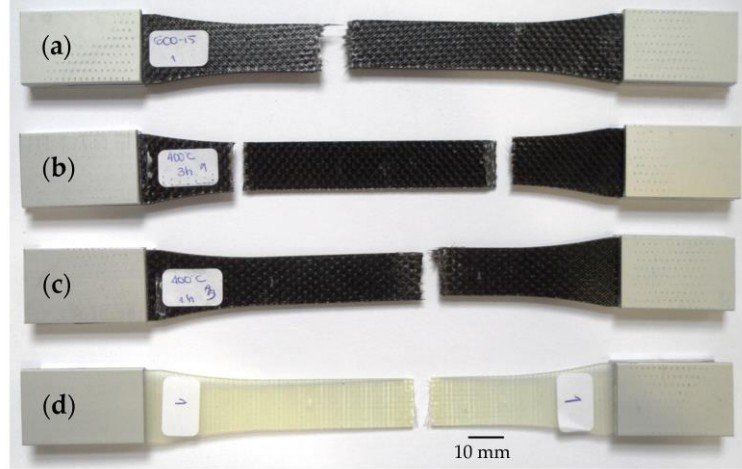

**Figure 4.** Examples of different samples tested: (**a**) 15 min at 600 °C; (**b**) 240 min at 400 °C; (**c**) 60 min at 400 °C; (**d**) control.

The tensile properties and representative tensile stress–strain curves are shown in Figure 5a. These curves are representative of all of them for the same conditions and are characterised by a linear increase in tensile stress with strain (linear elastic region) almost to the maximum bending load, followed by an abrupt decrease due to the impending collapse of the laminates.

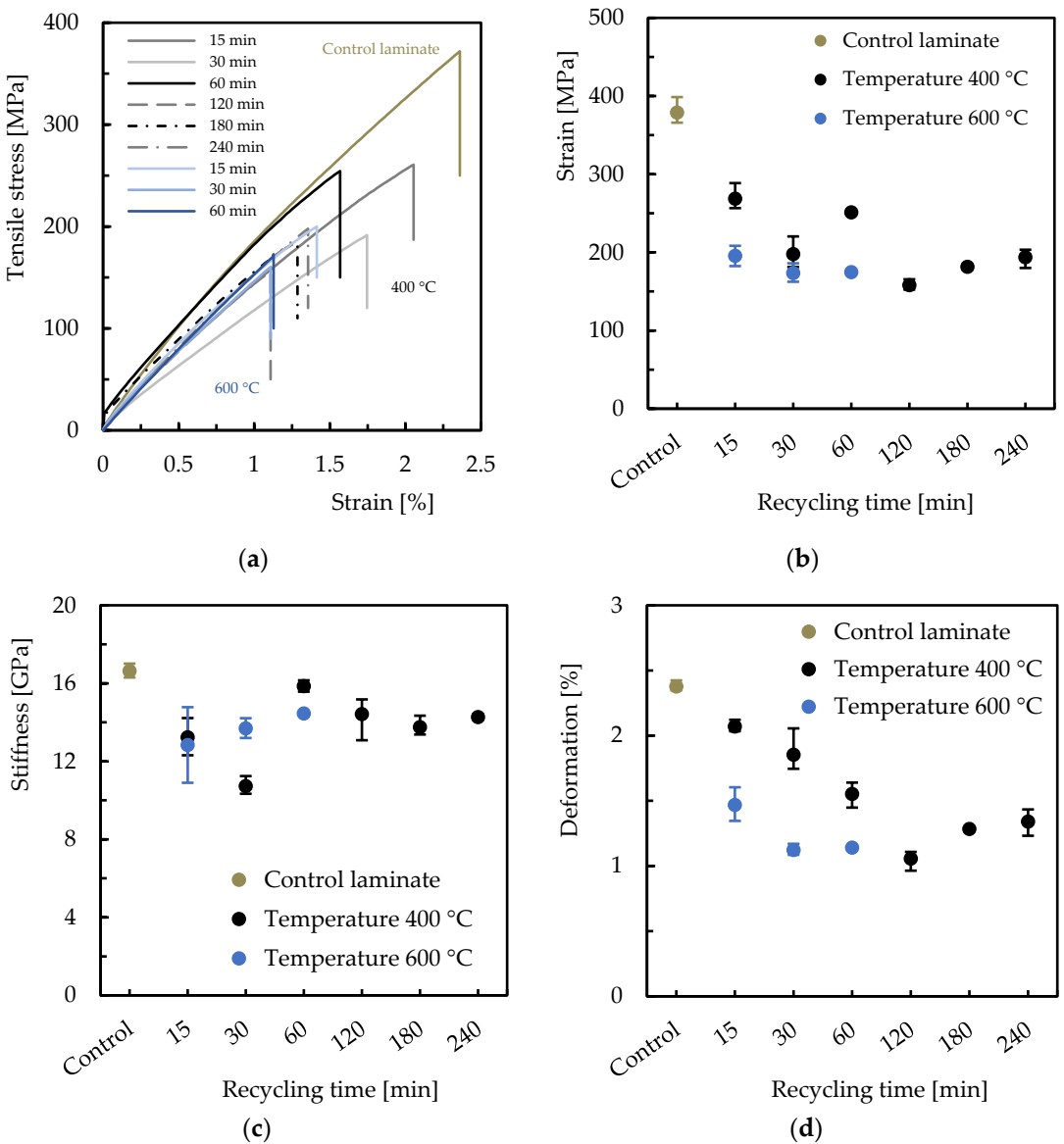

**Figure 5.** Tensile mechanical characterisation of fibre glass control laminate and laminates manufactured from the recycled fibre at a temperature of 400 °C and 600 °C and different recycling times: (**a**) representative tensile stress-strain curves; (**b**) tensile stress; (**c**) stiffness; (**d**) strain.

Regardless of the recycling conditions, the various representative curves are similar to the control laminate curve, where the influence of temperature and recycling time on the properties of the laminates is notorious [29].

Similar to the previous results, the lower temperature promotes better tensile properties, and as the time increases, the values of the tensile stress and deformation tend to decrease; on the other hand, as the recycling time increases for both temperatures, the stiffness value tends to increase. Having as reference the recycling time of 60 min, in relation to the control laminate and for the temperature of 400 °C, there is a reduction in terms of tensile tress of 33.7%, stiffness of 4.7%, and deformation of 34.7% (Figure 4b–d).

### 3.3. Flexural Properties

3PB tests were performed in order to understand the temperature and time effect on the bending properties of the laminates manufactured with the various recycled glass fibres. Figure 6 shows the main bending properties. Fibre breakage on the tensile side is the main damage mechanism observed, and delamination appears at the fibre–matrix interfaces, while the crack propagates through the cross-section of the laminate. In fact, the literature reports that this failure mode is typical of composites involving glass fibres because these fibres have low tensile strength. In addition, the high compressive stress concentration in the pin load contact region must also be considered [26,30,31]. The main failure modes in the control laminate includes fibre breakage, matrix cracking, and fibre–matrix debonding, as shown in Figure 6a. In the laminates of recycled fibres at a temperature of 400 °C for 180 min, we observed fibre–matrix debonding in the outer layers, fibres subject to traction and compression, and fibres breakage (Figure 6b). Finally, in one of the most critical cases, namely recycled fibres at a temperature of 600 °C for 60 min, all the damage mentioned above along the entire cross-section of the laminate is visible, as given in Figure 6c. Another conclusion was found: as the recycling time increases, the adherence of the epoxy resin to the fibres decreased, supporting that for higher times, lower mechanical properties result.

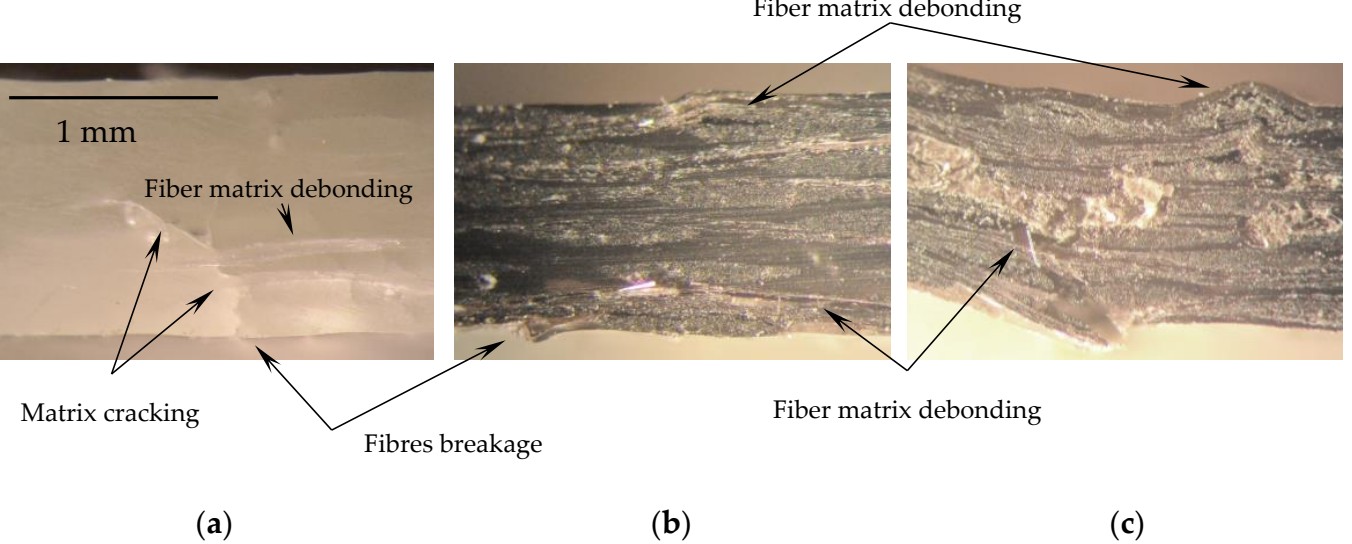

**Figure 6.** Damage mechanisms for laminates: (**a**) control; (**b**) 180 min at 400 °C; (**c**) 60 min at 600 °C.

Bending stress-strain curves are shown in Figure 7a for all laminates in the study and compared with the glass fibre control laminates. These curves are characterised by a linear increase in bending stress with strain (linear elastic region) almost to the maximum bending load, followed by a decrease due to the impending collapse of the fibres. The very evident zigzag appearance in the curves of the recycled fibre laminates results from the progressive break of the fibres.

Observing Figure 7b, the decrease of the bending stress of the laminates for both recycling temperatures is remarkable; for example, for the recycling time of 60 min the reduction is 54% and 57.9% for the temperatures of 400 °C and 600 °C, respectively, when compared with the control laminate. In terms of bending stiffness, and for a recycling time of 60 min, the reduction is 25% and 14.9% for the temperatures of 400 °C and 600 °C, respectively. Regarding bending strain (Figure 7d), at both temperatures, as the recycle time increases, the strain values decrease.

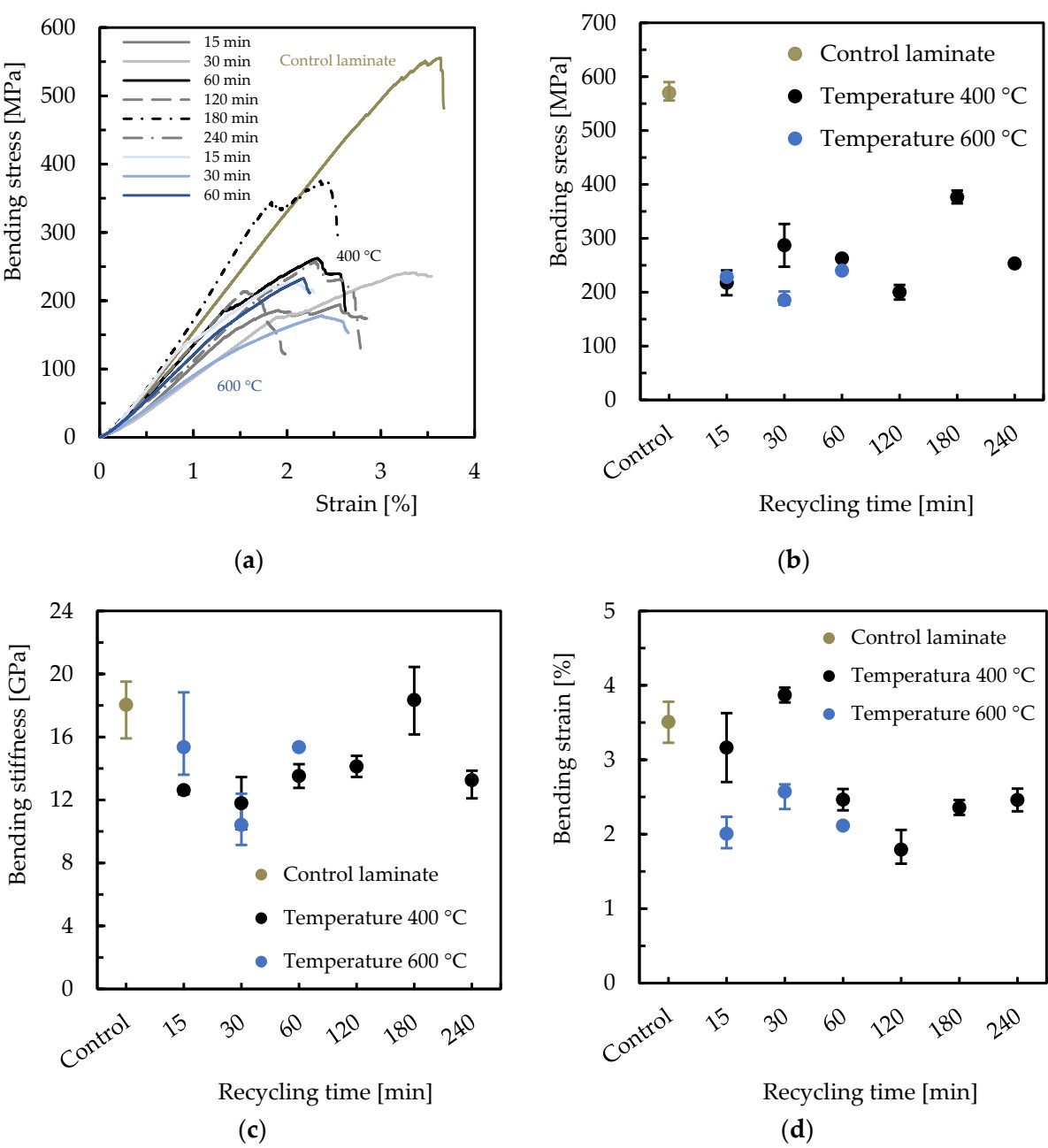

**Figure 7.** Flexural mechanical characterisation of control laminate and laminates manufactured with recycled glass fibres at a temperature of 400 °C and 600 °C and different times: (**a**) representative bending stress–strain curves; (**b**) bending stress; (**c**) bending stiffness; (**d**) bending strain.

### 3.4. Viscoelastic Properties

Figure 8 shows the strain rate effect on the bending properties for control glass fibre laminates. For all strain rates, regardless of the value, the behaviour is similar to those described in Figure 7a and in line with the open bibliography [26]. The main flexural properties at different strain rates (bending stress, bending stress, and bending strain) are obtained from the stress–strain curves shown in Figure 8, and they are analysed as shown in Figures 9 and 10 for temperatures of 400 °C and 600 °C, respectively. In this context, such properties are plotted against the logarithm of strain rate (log $\dot{e}$), where the symbols represent the average values.

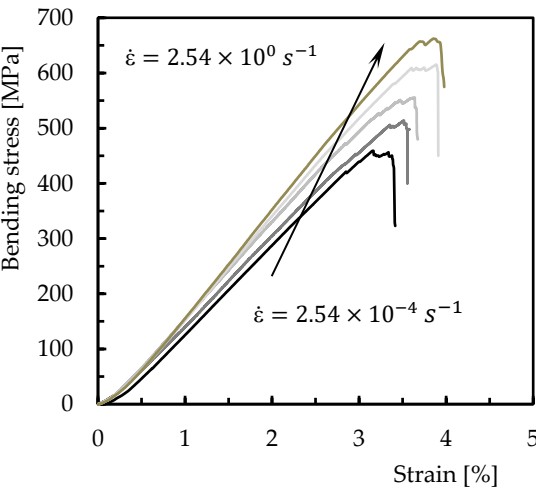

**Figure 8.** Representative bending stress–strain curves for glass fibre control laminate of all strain rates.

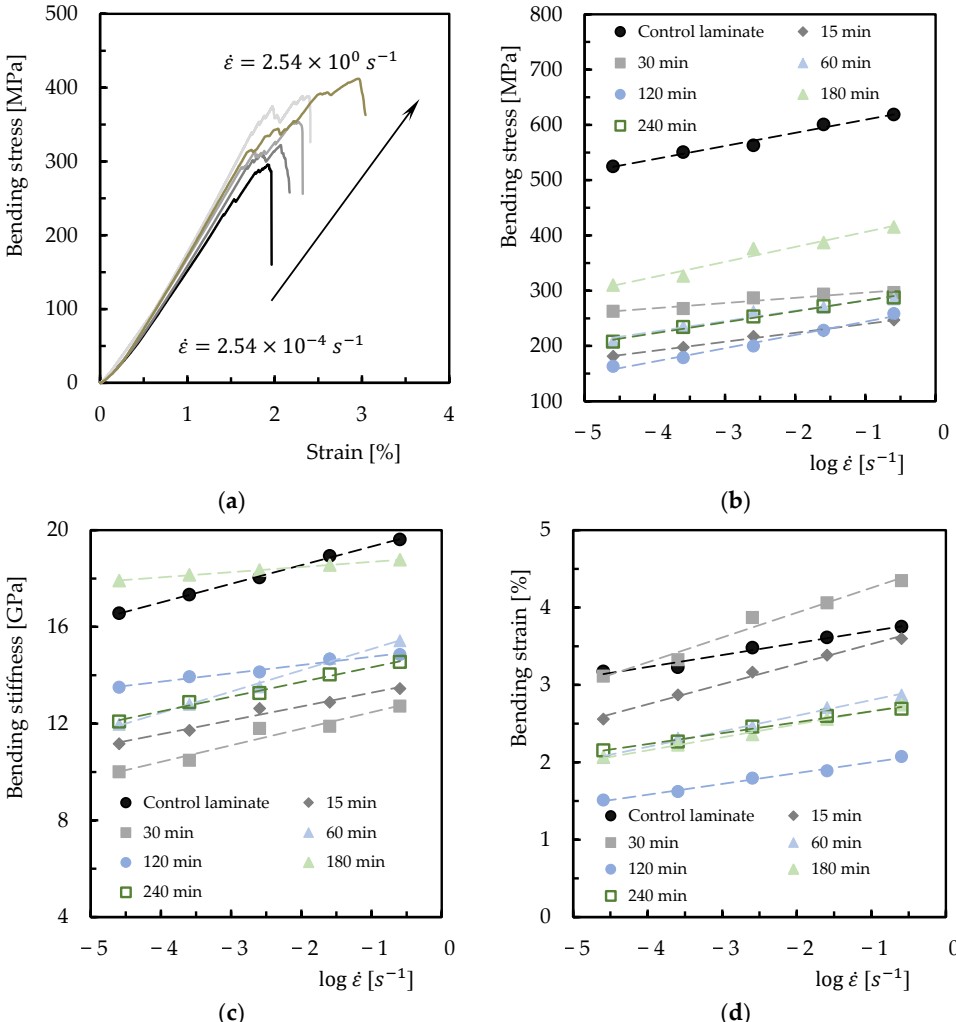

**Figure 9.** (**a**) Representative bending stress–strain curves for control laminate and laminates manufactured with glass fibre recycled at a temperature of 400 °C for 180 min of all strain rates; effect of the strain rate on the (**b**) bending stress; (**c**) bending stiffness; (**d**) bending strain.

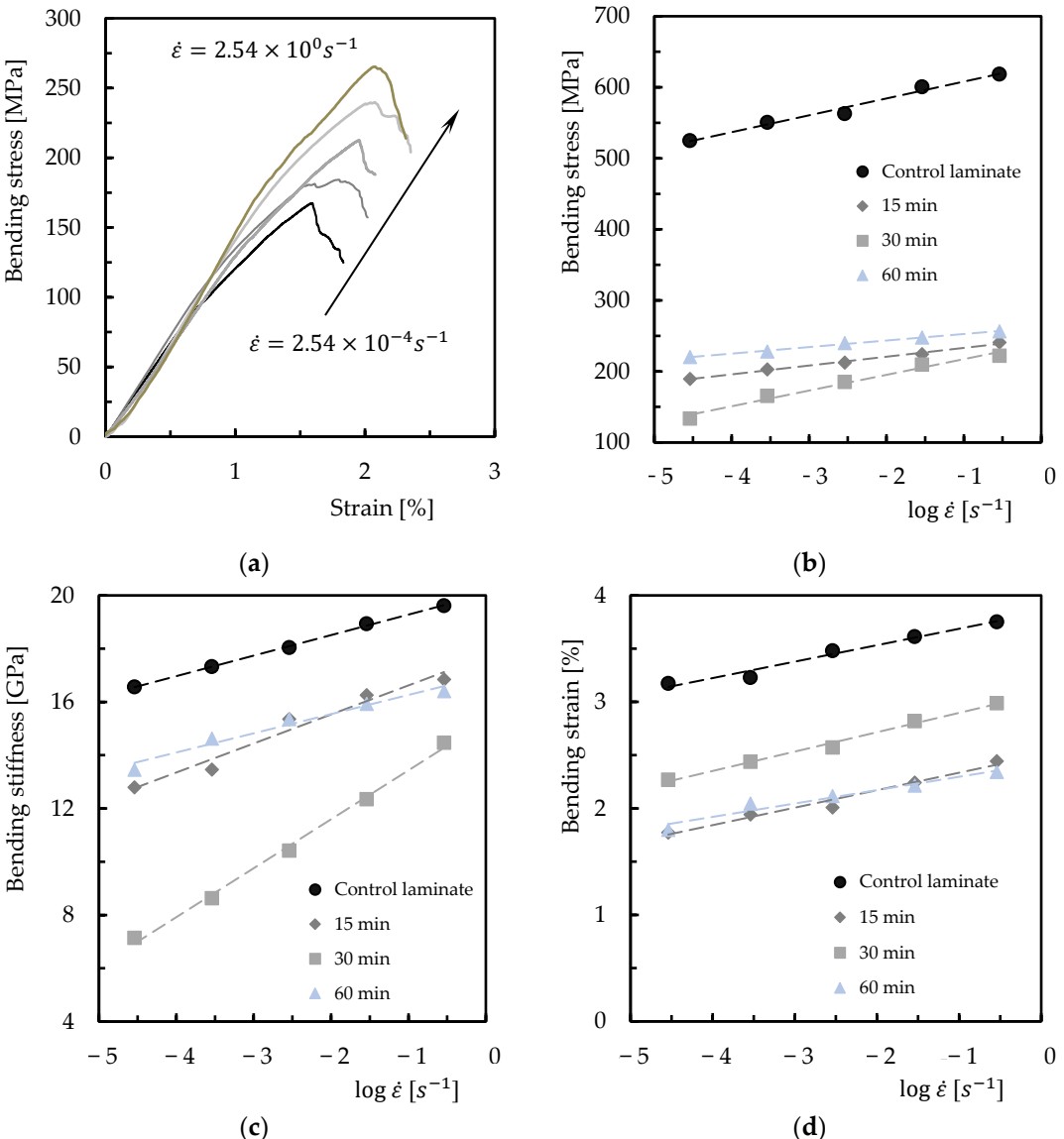

**Figure 10.** (**a**) Representative bending stress–strain curves for control laminate and laminates manufactured with glass fibre recycled at a temperature of 600 °C for 60 min of all strain rates; effect of the strain rate on the (**b**) bending stress; (**c**) bending stiffness; (**d**) bending strain.

Figure 9a shows the representative curves of the laminate with the best static properties for recycling at a temperature of 400 °C for 180 min, and it was possible to observe a tendency of growth of the value of the maximum bending stress with the increase of the strain rate, similar to that observed in the control laminate.

Figure 9b shows the evolution of the tension with the logarithm of the strain rate of the laminates manufactured with all the fibres recycled at a temperature of 400 °C in comparison with the control laminate (black dots). We thus note the trend obtained with the laminate of fibres recovered for 180 min, which showed greater similarity with the control laminate. In terms of bending stiffness, Figure 9c shows the trends obtained for the same laminates. For the laminate of the recovered fibres at 180 min and for low strain rates, the stiffness values were higher than the values of the control laminate. Figure 9d presents the results obtained from the various laminates for the effect of the logarithm of the strain rate on the bending strain by the laminates.

For manufactured laminates with recovered fibres at 600 °C, Figure 10a shows as an example the strain rate effect on the flexural properties of the laminate manufactured

with recovered fibres for 60 min. Figure 10b–d show the evolution of bending stress, stiffness, and strain, respectively, with the logarithm of the strain rate. Of the three laminates under study for the temperature of 600 °C, the laminate with the fibres recycled at 60 min is the one that stands out despite the substantial loss of properties.

In terms of viscoelastic behaviour, Figure 11 presents the stress–relaxation curves for the control laminate and the laminates with better static properties for the two temperatures in the study. This figure plots the average bending stress versus time. A fixed displacement was applied, with values corresponding to bending stresses of 285 MPa, 188 MPa, and 120 MPa for control laminate and recycled glass fibre laminates during 180 min at 400 °C and 60 min at 600 °C, respectively, that correspond to 50% of the maximum bending stress of each laminate.

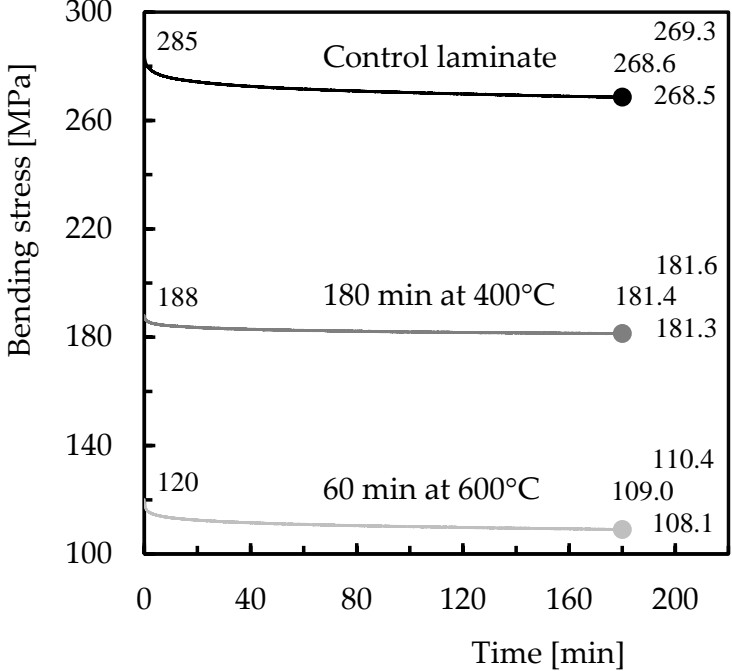

**Figure 11.** Stress–relaxation curves for best configurations studied.

For all laminates, we noticed a decrease in the stress from an initial value to one that is not yet constant because this study focuses on short-term tests of 180 min. These tests are an easy, fast, and reliable method to predict long-term behaviour [26,28], where this constant value will be expected only for higher stress values or longer periods of time. Another piece of evidence conveniently reported in the open literature is the existence of an initial regime in which the stress decreases considerably in relation to the remaining time [27]. Stress decrease for this material is especially important for the first 10 min. The behaviour of the laminates manufactured with recycling fibres is similar to that of the control laminate.

In terms of creep behaviour, Figure 12 shows typical curves obtained from the experimental tests. For this study, the same fixed bending stresses were applied as for the relaxation study. It is possible to observe that all curves present an instantaneous displacement, followed by the primary and secondary creep regimes that characterise the typical creep curves. In the present study, the third regime was expected to occur only for higher stress values or longer times. In this case, the displacement for this material is especially important for the first 30 min. The behaviour of the laminates manufactured with recycling fibres is also similar to that of the control laminate although their response is lower.

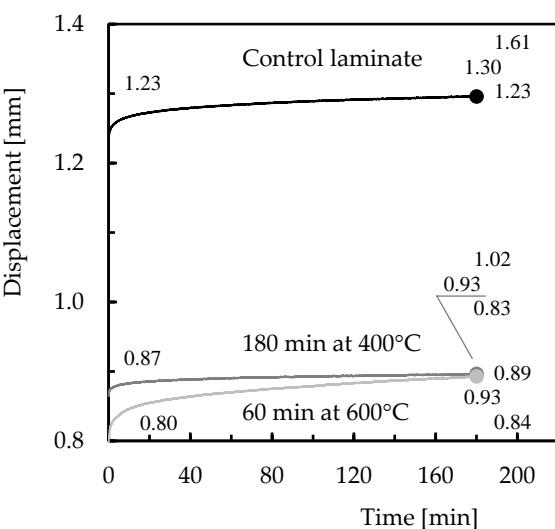

**Figure 12.** Creep behaviour for best configurations studied.

## 4. Discussion

As for the analysis and discussion of results, we intended to find out which of the nine time/temperature relationships has the most similar properties to those of control, that is, to find out the ideal temperature and recycling time. For this, the mechanical characterisation shown, the microscopic evaluation, as well as the EDX evaluation of the different samples all contribute.

In the study carried out on glass fibre strands, shown in Figure 2, the control sample has an average value of 34.6 (±2.2) N. In samples tested at 400 °C, the average value varies between 14.5 (±0.8) N and 16.3 (±2.7) N for 15 and 240 min, respectively. As for tensile strength compared to the control sample, these samples show a decrease in tensile strength compared to the control, which ranges from 52.9% to 58.1%. It should be noted that for a temperature of 400 °C, there is not a large variation in tensile strength between 120 min and 180 min of recycling; therefore, it is not necessary to subject the samples to a longer burning time, as they will tend to degrade. On the other hand, from the point of view of an industrial application, it would not be necessary to exceed the times tested, as it would not represent an enhancement of the properties of the laminate, and it would increase the costs of the recycling process. For samples subjected to a fixed temperature of 600 °C, once again varying the recycling time, the average value varies between 9.7 (±1.5) and 12.7 (±3.5) N. For this temperature, the samples exposed for 15 min 30 min and 60 min show a decrease of 63.3%, 70.5%, and 71.7% compared to the control, respectively.

It is evident that there is a tendency for the tensile strength to decrease with increasing temperature, as observed by Beauson et al. [32], with different temperatures and firing times, namely 350 °C for 300 min and 400 °C for 120 min, as the authors observed a reduction of 38.2% and 58.5% in tensile strength compared to the control, respectively. When compared with the samples under study, for the same temperature and the same burning time, 400 °C for 120 min, the value obtained was lower, i.e., 56.9%, in relation to the referred study. Moreover, Oliveux et al. [14] reported that the mechanical properties of glass fibres decrease by at least 50% for temperatures between 500 °C and 550 °C.

By analysing the tensile tests on the bidirectional woven glass fibre (Figure 3), it was verified that the control sample presents an average value of 350.3 (±3.5) N. For the samples of 400 °C, and varying the recycling time, the average value varies between 180.1 (±8.8) N and 712.3 (±15.8) N. As for the samples subjected to recycling times of 15 min, 30 min, and 60 min, the value of their resistance to traction increased compared to the control, namely by 103.3%, 95%, and 22.7%, respectively. This can be explained by the conditions to which they were subjected; that is, these samples were not subjected to an ideal recycling time since, for these times, the woven glass is not completely clean, showing residues of

epoxy resin. That is, the load is distributed among the fibres, resulting in an increase in tensile strength for these samples. That said, it is assumed that these temperatures and burning times are ineffective for the intended purpose, making these samples invalid in this study. For the same temperature and varying only the recycling time, namely to 120 min, 180 min, and 240 min, the samples present an average value between 180.1 ($\pm$8.8) N and 281.2 ($\pm$12.2) N. For the samples that were recycled at 400 °C for 15 min, 30 min, and 60 min, their average value increased by 103.3%, 95.0%, and 22.7% compared to the control. For higher values of recycling time, i.e., 120 min, 180 min, and 240 min, there is a decrease in the mean value of 19.7%, 28.9%, and 48.6% compared to the control, showing woven fibre degradation.

Regarding the samples recycled at 600 °C for 15 min, 30 min, and 60 min, the mean value is 296.8 ($\pm$7.7) N, 243.7 ($\pm$10.8) N, and 141.1 ($\pm$4.7) N, showing a decrease in tensile strength of 15.3%, 30.4%, and 59.7%. As observed in relation to glass fibre strands and also for fabric, as the temperature and burning time increase, the tensile strength decreases. Beauson et al. [32] also confirmed this trend, which can be explained by the methodology adopted since it uses very high temperatures.

A trinocular stereoscopic microscope (TSM) from Nikon, model SMZ-2t, was used with a 10× objective with SEM images at a magnification of 200×, 1000×, and finally 2000× to obtain a more detailed description of the studied samples. In Tables 2 and 3, the images relating to the different samples under study are presented, with the main objective of the microscopic evaluation being to confirm and analyse in a concrete way the results obtained so far.

In the sample of control fibres, the presence of some impurities on their surface can be noted, as can be seen in Table 2 (electromagnetic lens with a magnification of 200 times). Despite the presence of impurities, it presents a smooth and uniform surface. Rahimizadeh et al. [19] found unequal impurities in the control glass fibres and also described the surface of the control glass fibre as smooth and homogeneous.

The presence of epoxy resin can be observed, highlighted with the orange arrows, in samples submitted to 400 °C for 15 min, 30 min, and 60 min (Table 2). Beauson et al. [31] faced the same consequence, as the recycling did not take place entirely, with the fibres having a black appearance. After analysing the results obtained in the tensile tests, it was noticed that these three samples are the ones that present a superior tensile strength compared to the control sample, namely, 103.3%, 95%, and 22.7%. These are coated with resin particles, in which the load is distributed among all the fibres, leading to an increase in tensile strength. According to authors Krishna et al. [32], the tensile strength of the fibres varies with the amount of resin, revealing that the samples that recorded high amounts of resin had the highest values of tensile strength. Relative to the remaining samples subjected to the temperature of 400 °C and recycling times of 120 min, 180 min, and 240 min, as mentioned before, these samples show a decrease in tensile strength. As can be seen in Table 2, there are some fibre breaks. Regarding the fibres submitted to a temperature of 600 °C and recycling times of 15 min, 30 min, and 60 min, there is also a decrease in their tensile strength compared to the control because, as we can see in Table 3, on the surface, there is debris resulting from the recycling process, or for longer times, they suffer breakage and consequent degradation.

As also analysed by Beauson et al. [33], the glass fibre samples that have a clean appearance are those that show a lower tensile strength.

Finally, an EDX evaluation was carried out in order to quantify the main chemical elements of glass fibres after recycling. For this, in order to have a comparative term, the EDX evaluation was carried out at different temperatures for the lowest and highest recycling times (Figure 13).

**Table 2.** TSM and SEM images of the glass fibre control and glass fibres after recycling at a temperature of 400 °C and at different times.

| Samples | 10× Objective | 200× Magnification | 1000× Magnification | 2000× Magnification |
|---|---|---|---|---|
| Control | | | | |
| 15 min | | | | |
| 30 min | | | | |
| 60 min | | | | |
| 120 min | | | | |
| 180 min | | | | |
| 240 min | | | | |

**Table 3.** TSM and SEM images of the glass fibres after recycling at a temperature of 600 °C and at different times.

| Samples | 10× Objective | 200× Magnification | 1000× Magnification | 2000× Magnification |
|---|---|---|---|---|
| 15 min | 1 mm | 200 µm | 50 µm | 20 µm |
| 30 min | 1 mm | 200 µm | 50 µm | 20 µm |
| 60 min | 1 mm | 200 µm | 50 µm | 20 µm |

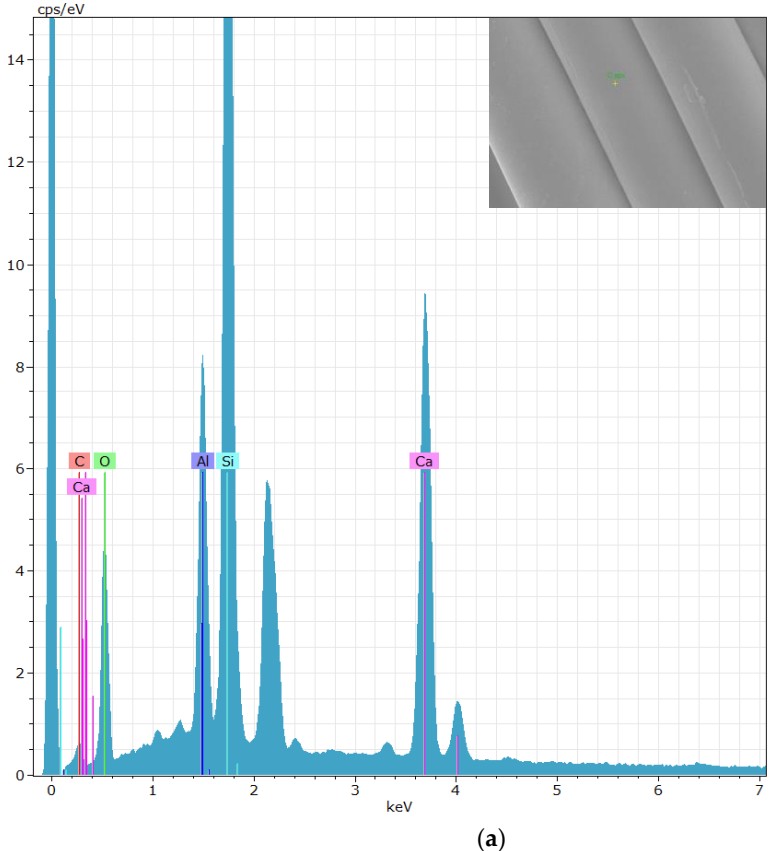

(**a**)

**Figure 13.** *Cont.*

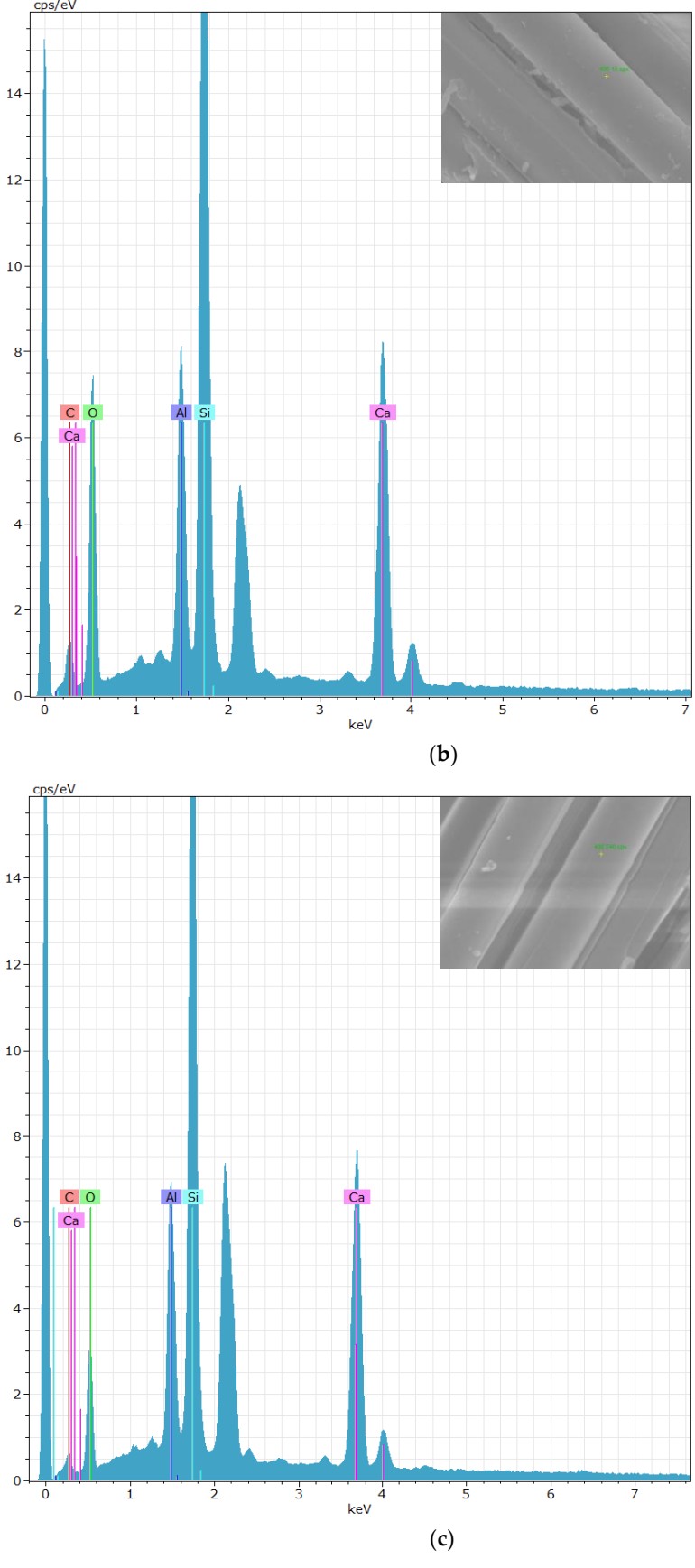

**Figure 13.** *Cont.*

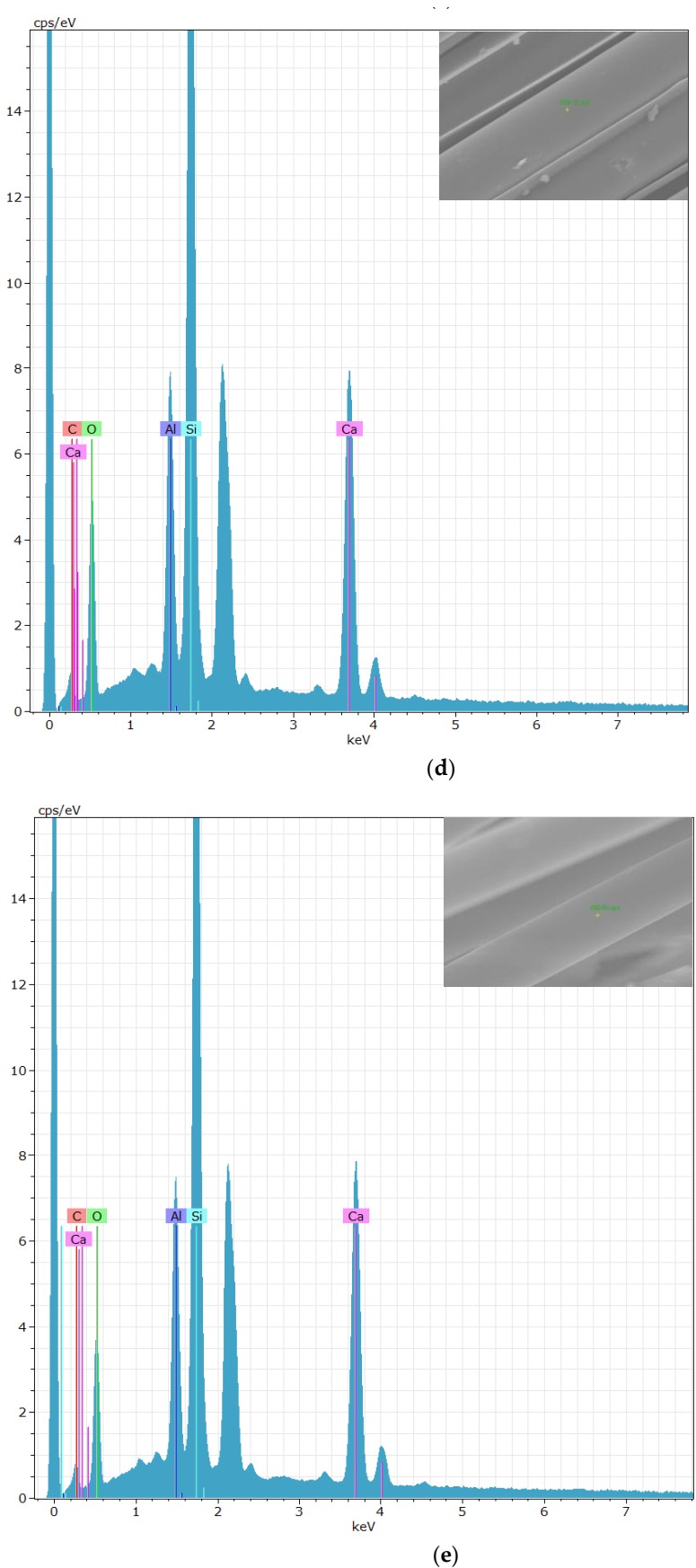

**Figure 13.** Scattered energy spectrum, referring to the sample of (**a**) control; (**b**) 400 °C during 15 min; (**c**) 400 °C during 240 min; (**d**) 600 °C during 15 min; (**e**) 600 °C during 60 min.

It is important to point out that the type of fibre used has a silane treatment; therefore, the different chemical elements in its constitution, also emphasise that, according to Oliveux et al. [14], the resin, when volatilised, produces some gases (carbon dioxide ($CO_2$), hydrogen ($H_2$), and methane ($CH_4$)) and oil fractions. In Table 4, the mass percentage of each chemical element constituting the different samples of glass fibres subjected to different temperatures, for the lowest and highest burning times, is presented, with a special focus on the chemical elements of carbon and silica.

**Table 4.** Chemical elements of the samples.

| Laminate | Mass Percentage (%) | | | | |
|---|---|---|---|---|---|
| | Carbon (C) | Oxygen (O) | Aluminium (Al) | Silica (Si) | Calcium (Ca) |
| Control | 8.33 | 34.46 | 8.66 | 26.27 | 22.28 |
| 400 °C during 15 min | 14.34 | 41.08 | 7.48 | 21.11 | 15.98 |
| 400 °C during 240 min | 11.46 | 30.4 | 8.8 | 26.89 | 22.45 |
| 600 °C during 15 min | 12.29 | 35.58 | 8.37 | 24.4 | 19.37 |
| 600 °C during 60 min | 12.72 | 35.64 | 8.97 | 26.23 | 26.29 |

Comparing the sample subjected to a temperature of 400 °C for 15 min with the control sample, there is a significant increase in the percentage by mass of C and a decrease in the percentage by mass of Si. For the sample at 400 °C for 240 min, there is a decrease in the C mass percentage and an increase in the Si mass percentage. Comparing both samples, the one that was subjected to the longest burning time has a lower percentage of C, while the one that was subjected to the shortest recycling time has a higher percentage of C.

When subjected to a temperature of 400 °C for 15 min, the increase in C occurs due to the presence of resin, as shown in Table 2 above, remains high, which can be explained by the increase in coal content. These results can be corroborated by the studies developed by Gopalraj et al. [24].

As for the sample subjected to 600 °C for 15 min, it shows a decrease in Si compared to the control. In relation to the longer recycling time, an increase in the mass percentage of C of 0.5% is revealed, contributing to the decrease in tensile strength by 44.4%. It is concluded that, for both samples, there is a decrease in the Si content and an increase in the C content when compared with the control. This can be justified in the case of the sample subject to the shortest burning time, i.e., 15 min, by the presence of resin and in the sample subject to the longest burning time, i.e., 60 min, to the formation of charcoal on the surface of the fibres, as verified by Gopalraj et al. [24].

Regarding the tensile characterisation of the control laminates, they showed an average maximum stress of about 378.84 (±14.21) MPa and a stiffness of 16.63 (±0.29) GPa (Figure 5). For the laminates whose fibres were subjected to a temperature of 400 °C, there was a decrease of 30% in the maximum tension, reaching an average value of 268.64 (±14.22) MPa, and a decrease of about 20% in the stiffness, reaching a value of 13.23 (±0.78) GPa for the laminates that have the best properties and that of fibres for 15 min.

In the case of the maximum tension reached by the recycled fibre laminates at 600 °C, the best result was obtained by the 15 min fibres, with an average value of about 195 (±) MPa, which represents a decrease of about 48% when compared to the value of the control laminate. These laminates show very small variations between the different recycling times, as can be seen in Figure 5, and for both of the times of 30 min and 60 min, there was a decrease of 54% and 55.4% compared to the tension of the control laminate. As for stiffness, it increases with the increase in fibre recycling time, and the laminate of fibres recovered after 60 min has a stiffness of 14.45 (±0.41) GPa, which represents a 23% reduction compared to the control laminate.

In terms of static characterisation of 3PB tests, as given in Figure 7, the control laminate obtained a maximum tension of 560 (±20.18) MPa and a stiffness of 18.04 (±1.33) GPa, showing that, once again, the dispersion of the trials is not significant.

The laminate of fibres recycled at a temperature of 400 °C for 180 min stands out from the rest, presenting a maximum tension of 376.24 ($\pm$9.78) MPa. In relation to the control laminate, there was a 34% reduction in tension and an increase of 1.7% in stiffness.

For the temperature of 600 °C, the laminates recovered for 15 and 60 min show the best flexural properties, with stresses of 228.14 ($\pm$5.17) and 239.98 ($\pm$11.42) MPa, respectively, and stiffness of 15.36 ($\pm$1.54) and 15.36 ($\pm$1.42) MPa, respectively. Comparing with the control laminate, for the laminate with fibres recovered during 15 min, in terms of tension and stiffness, there was a reduction of 149.8 % and 17.4 %, respectively, and for the 60 min laminate, a reduction of 137.5 and 17.5 % in terms of tension and rigidity was observed.

In terms of strain rate and independent of the laminate, higher strain rates promote higher maximum bending stresses. In terms of control laminate, for example, Figure 8 evidences that the maximum bending stress increases from 524.7 MPa at $2.54 \times 10^{-4}$ s$^{-1}$ to 618.4 MPa at $1.3 \times 10$ s$^{-1}$, which represents an increase around 17.9%. The bending modulus increases from 16.6 GPa to 19.6 GPa, which represents an increase around 18.1%.

Figure 9a shows the representative curves of the laminate that presented the best characteristics for recycling at a temperature of 400 °C for 180 min, and it is possible to observe that there is a tendency to increase the maximum stress reached with the increase in the strain rate, similar to that observed in the control laminate. For these conditions of recycling fibres, in terms of stress, the values increase from 310.3 MPa to 415.3 MPa, which is an increase of 33.8%, and in terms of stiffness an increase of 5.0%, i.e., from 17.9 GPa to 18.8 GPa. Figure 9b,c shows the evolution of stress with the logarithm of the strain rate and the trend of stiffness as a function of the logarithm of the strain rate. These trends become even more evident, and of the six laminates studied, the fibre laminate recycled for 180 min was the one that showed the greatest similarity compared to the control laminate. This behaviour translates into a parallel linear trend regarding stress, with the decrease compared to the control laminate being around 1.8 times greater. In terms of deformation (Figure 9d), the laminate of fibres recovered during 180 min presented a deformation that, although lower than the control one, was parallel to this. Here, the fibre laminate recovered for 30 min also stands out, as it presents a higher deformation than the control laminate in practically all the deformation rates considered.

Figure 10a shows the representative curves of the laminate that presented the best properties for recycling at a temperature of 600 °C for 60 min, where the already-reported trends are observable. For stress, there is a growth of 16.4%, i.e., from 220.2 MPa to 256.4 MPa. In terms of stiffness, there was an increase of 21.5%, i.e., from 13.5 GPa to 16.4 GPa, and for strain, we observed an increase of 28.5. For this temperature, the behaviour of the remaining laminates for 15 min and 30 min is similar.

Regardless of the trend that laminates present with regard to the evaluated mechanical properties, as suggested by the literature [33–37], a linear model can be fitted to the data according to the following equations:

$$s = a + b \times \dot{e} \tag{7}$$

$$E = a + b \times \dot{e} \tag{8}$$

$$\varepsilon = a + b \times \dot{e} \tag{9}$$

where $s$ is the maximum bending stress, $E$ is the bending modulus, $\varepsilon$ is the strain at maximum bending stress, and $\dot{e}$ is the logarithm of strain rate, and $a$ and $b$ constants are presented in Table 5. Therefore, from this table, it is possible to observe that the proposed linear relationships (between logarithm of strain rate and mechanical properties) present good precision (R greater than 0.95) and can be used as prediction tools.

**Table 5.** Parameters of the equations that fit the effect of the strain rate.

| Laminate | Properties | Parameters | | |
|---|---|---|---|---|
| | | *a* | *b* | R |
| Control | Bending stress ($\sigma$) | 23.73 | 633.0 | 0.9901 |
| | Bending modulus (*E*) | 0.77 | 20.1 | 0.9993 |
| | Flexural modulus ($\varepsilon$) | 0.15 | 3.8 | 0.9863 |
| 400 °C during 180 min | Bending stress ($\sigma$) | 27.04 | 433.3 | 0.9816 |
| | Bending modulus (*E*) | 0.21 | 18.9 | 0.9996 |
| | Flexural modulus ($\varepsilon$) | 0.17 | 2.8 | 0.9975 |
| 600 °C during 60 min | Bending stress ($\sigma$) | 9.21 | 261.9 | 0.9976 |
| | Bending modulus (*E*) | 0.72 | 17.0 | 0.9829 |
| | Flexural modulus ($\varepsilon$) | 0.13 | 2.4 | 0.9756 |

In order to better understand the viscoelastic behaviour of the studied laminates, a study was carried out in terms of creep and stress relaxation of the laminates that presented the best properties in the static 3PB tests, allowing us to know the dimensional stability and its long-term resistance.

Figure 11 presents the stress–relaxation curves of the control laminates and laminates with recycled fibres at 400 °C and 600 °C for 180 min and 60 min, respectively. For the control laminate, a decrease in tension of about 5.84% is visible compared to the initial value of 285.0 MPa, reaching 268.6 MPa after 180 min. It should be noted that there is a more pronounced relaxation in the first 30 min, representing about 4.22%, while for the remaining 150 min, the reduction is only 1.70%. Similar behaviour was observed in the remaining laminates under study. In the laminate whose fibre-recycling temperature was 400 °C, with an initial stress of 188.0 MPa, it suffered a relaxation of 3.5%, reaching a final value of 181.4 MPa. With the control laminate, at 30 min, the highest relaxation value had already occurred, with a 2.62% decrease in tension, while during the rest of the test, the reduction was 0.96%. The laminate manufactured from recycled fibres at 600 °C was the one with the greatest stress relaxation since, for an initial stress of 120 MPa, it suffered a decrease of 9.21% at 180 min, translating into a final stress of 109.0 MPa. Additionally, analysing the first 30 min, there was a reduction of 6.82%, followed by one of 2.56% in the remaining 150 min.

Concerning the creep behaviour, Figure 12 shows typical curves obtained from the experimental tests of the control laminates and laminates with recycled fibres at 400 °C and 600 °C for 180 min and 60 min, respectively. For the control laminate, an increase in displacement of 5.26% is visible compared to the initial value of 1.23 mm, reaching approximately 1.30 mm at 180 min. It is possible to notice that a more accentuated deformation occurs in the first 30 min representing 3.71% and a deformation of 1.5% in the remaining 150 min. The creep of the recycled fibre laminate during 180 min at 400 °C presented an initial deformation of 0.87 mm, suffered a displacement corresponding to 3.57%, and reached a final value of about 0.93 mm. As with the control laminate, at 30 min, the largest portion of displacement had already occurred, with a deformation of 2.50%. Finally, the laminate made from recycled fibres at 600 °C for 60 min presented the greatest deformation since, with an initial displacement of 0.80 mm, it suffered an increase of 12.26% after 180 min, translating into a displacement of 0.93 mm Moreover, analysing the first 30 min, there is an increase of 8.12%.

## 5. Conclusions

In view of the initially defined objectives, it is possible to draw some conclusions after analysis and discussion of results regarding laboratory practice, namely tensile tests and 3PB tests, viscoelastic behaviour, microscopic evaluation, and EDX evaluation.

There is greater tensile strength for samples exposed simultaneously to lower temperature and shorter recycling times.

In general, the increase in temperature and, simultaneously, the increase in recycling time causes a loss of tensile strength in relation to the control fibreglass. For the glass fibre strands, the best recycling condition entails a temperature of 400 °C for 60 min, with the lowest tensile strength compared to the control 52.9%. As for the woven glass fibre, the best recycling condition among those studied is at a temperature of 600 °C for 15 min since it is the one that presents results closest to those of the control: a loss of tensile strength of 15.3% compared to the average control.

From a microscopic evaluation, it was inferred that for the high values of tensile strength, there was a greater amount of resin on the surface of the glass fibres. Therefore, it is concluded that these samples were not subjected to ideal temperatures/burning times. On the other hand, smaller amounts of resin or even its absence were noted in the samples that exhibited the lowest values of tensile strength and presented a brittle behaviour.

For the higher temperature, there was a considerable increase in the percentage of C in the glass fibres, emphasizing that the higher the percentage of charcoal, the lower the tensile strength of the glass fibres.

Tensile laminate tests allow the conclusion that with the increase in fibre recycling time and consequent degradation of the same, there is a decrease in their tensile strength. One of the laminates with the best tensile properties is the one manufactured with recycled fibres at 400 °C for 60 min, with a reduction compared to the control of 5.2% and 4.4% in terms of tensile stress and stiffness, respectively. In the 3PB tests, the laminate of recycled fibres at 400 °C for 180 min stands out from the others because it presents the smallest reduction in percentage terms, that is, a 34% reduction in tension and an increase of 1.7% in stiffness.

The study of the viscoelastic behaviour of laminates manufactured from recycled fibres through by-hand lay-up technique allows us to understand their behaviour at different deformation rates that prove to be similar but, in most cases, with lower values than the control laminate. Even at room temperature and regardless of fibre type, stress–relaxation tests show that stress decreases over time. In terms of creep behaviour and regardless of fibre type, displacement increases with time.

Some of the resulting materials, as referred to above, present good mechanical properties and can, for example, be embedded in non-structural parts of cars as a way of reducing the material consumption associated with the automotive industry but also to answer the European Directive 2000/53/CE that defines 80% as the minimum amount of recycled materials in cars.

In future studies, it could be interesting to refine recycling temperatures and times but also try to comprehend the phenomenon associated with the laminates recycled for 180 min at 400 °C. The small oven used greatly impacted the ability to perform different tests, so it would also be very interesting to characterise the laminates, for example, through fatigue tests.

Considering future applications of this study, it would be interesting to perform a cost–benefit analysis to understand how it would impact the environment, study what it would represent in terms of the logistic aspect of the recovery of the materials for recycling, and determine the means necessary for it, aiming to understand the volumes of materials for recycling needed to make the recycling process viable on an industrial scale. In terms of studying the applicability of the results obtained here, the main interest would be to reproduce the procedure on material at the end of its useful life, such as wind turbines' rotor blades or parts used in the aeronautical and automotive industries.

**Author Contributions:** Conceptualization, T.M.L. and P.S.; methodology, T.M.L. and P.S.; validation, T.M.L. and P.S.; formal analysis, T.M.L.; investigation, T.M.L., P.S., M.I. and L.L.; resources, P.S.; data curation, M.I. and L.L.; writing—original draft preparation, M.I. and P.S.; writing—review and editing, T.M.L. and P.S.; visualization, P.S.; supervision, T.M.L. and P.S. All authors have read and agreed to the published version of the manuscript.

**Funding:** This research received no external funding.



**Institutional Review Board Statement:** Not applicable.

**Informed Consent Statement:** Not applicable.

**Data Availability Statement:** Not applicable.

**Acknowledgments:** The research work was supported in part by FCT—Fundação para a Ciência e a Tecnologia, I.P./MCTES, through national funds (PIDDAC), within the scope of the Unit I&D C-MAST (Center for Mechanical and Aerospace Science and Technology), Project UIDB/00151/2020.

**Conflicts of Interest:** The authors declare no conflict of interest.

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
