# Peer review of "Thermal Recycling of Glass Fibre Composites: A Circular Economy Approach"

_sustainability, doi:10.3390/su15021396_

Round 1

Reviewer 1 Report

This paper addresses an important topic of thermal recycling of glass fibre composites.

The title of the paper adds the following statement “ The challenge of the rotor blades in the end of their life cycle.” It is clear throughout the whole study that recycling the wind turbine blade is not the objective of this study. This title is miss leading. I request revisions to answer my following concerns:

(1)  Change the title to reflect the actual work

(2) Clearly state the objective of this paper; and show that results, discussion and conclusions are related to that objective

Author Response

We would like to thank you for carefully examining our work and for allowing us to revise and improve the manuscript. We have addressed all your comments and suggestions and modified the paper accordingly. All modifications are marked in blue colour in the revised manuscript to facilitate the review process.

Reviewer 2 Report

This is a good work for the thermal recycling of glass fibre composites. The article can be published after revising the following comments-

1. Please revise the abstract so that the overall findings and objectives of the work can be reflected properly. 

2. Introduction- author didn't provide any explanation - how Circular economy concept can be applied with the thermal recycling for composite industries. 

3. Literature review was very limited in terms of the recent work on thermal recycling of glass fibres. Author didn't describe any potential knowledge gap where the results of this work can contribute for further development in this field.

4. Likewise, the objectives of the work were not clearly mentioned in the introduction section. The objectives of the work should be clearly stated.

5. Materials and methods - why only three recycling time considered for 600 deg c. What is the thickness of the control laminate and volume fraction as well?

6. How author justifies the mechanical properties comparison between the 6 layers structure in recycled  laminate, while the control laminates used eight layers of fabric in the structure?

7. Please provide the test details carefully, such as tensile test- test speed, no. of repetations  etc.

8.  test methodology should have been described one by one. 

9. Why different strain rate bending test was done? What was the real life correlation for this test? Same for the creep test- please describe. Any test standard for the creep test?

10. Figure 2 and 3 - b, is it maximum load? Please check other figures accordingly as well.

11. Please point the  different damage mechanisms clearly in the picture.

12. Table 2- why 400 deg C 120 min sample showed similar type of characteristics in the picture as the control sample?

13. Indicate the epoxy resin for 400 deg c 15, 30 and 60 min samples in the pictures.

14. No condition was optimised ( time , temperature)  for the thermal  recycling, since it was an objective of this work.

15. Revise the conclusion- please make it clear how the outputs can be used in any other future work, how the results help in the real life application, any limitations of this work etc. 

Author Response

(The authors gave the same response as above.)

Reviewer 3 Report

1. Write more challenges on thermoset-based polymer composites.

2. What is life cycle?

3. Point out objective of the research.

4. Well written conclusion.

5. Mention significance of Viscoelastic properties.

Some specific questions to the authors

1. The conclusions may be improved to make consistent with the evidence and arguments presented and address the main question posted?
2. Add some more related references.
3. Include some additional comments on the figures 3, 4, 5, 6 and 7.

Author Response

(The authors gave the same response as above.)

Round 2

Reviewer 1 Report

The revised manuscript answers my concerns.